# Integrated Transcriptomic and Metabolomic Analyses Uncover the Differential Mechanism in Saline–Alkaline Tolerance between *Indica* and *Japonica* Rice at the Seedling Stage

**DOI:** 10.3390/ijms241512387

**Published:** 2023-08-03

**Authors:** Jianyong Wang, Keke Hu, Jien Wang, Ziyun Gong, Shuangmiao Li, Xiaoxiao Deng, Yangsheng Li

**Affiliations:** 1State Key Laboratory of Hybrid Rice, Key Laboratory for Research and Utilization of Heterosis in Indica Rice, Ministry of Agriculture, College of Life Sciences, Wuhan University, Wuhan 430072, China; wangjianyong90@sina.cn (J.W.); hukk126@126.com (K.H.); jienwang2014whu@163.com (J.W.); gziyun@whu.edu.cn (Z.G.); lishuangmiao@whu.edu.cn (S.L.); 2017102040003@whu.edu.cn (X.D.); 2Lushan Botanical Garden, Chinese Academy of Sciences, Jiujiang 332900, China

**Keywords:** rice (*Oryza sativa* L.), *indica*–*japonica* subspecies, saline–alkaline stress, transcriptome, metabolome

## Abstract

Saline–alkaline stress is one of the major damages that severely affects rice (*Oryza sativa* L.) growth and grain yield; however, the mechanism of the tolerance remains largely unknown in rice. Herein, we comparatively investigated the transcriptome and metabolome of two contrasting rice subspecies genotypes, Luohui 9 (abbreviation for Chao2R under study, *O. sativa* ssp. *indica*, saline–alkaline-sensitive) and RPY geng (*O. sativa* ssp. *japonica*, saline–alkaline-tolerant), to identify the main pathways and important factors related to saline–alkaline tolerance. Transcriptome analysis showed that 68 genes involved in fatty acid, amino acid (such as phenylalanine and tryptophan), phenylpropanoid biosynthesis, energy metabolism (such as Glycolysis and TCA cycle), as well as signal transduction (such as hormone and MAPK signaling) were identified to be specifically upregulated in RPY geng under saline–alkaline conditions, implying that a series of cascade changes from these genes promotes saline–alkaline stress tolerance. The transcriptome changes observed in RPY geng were in high accordance with the specifically accumulation of metabolites, consisting mainly of 14 phenolic acids, 8 alkaloids, and 19 lipids based on the combination analysis of transcriptome and metabolome. Moreover, some genes involved in signal transduction as hub genes, such as *PR5*, *FLS2*, *BRI1*, and *NAC*, may participate in the saline–alkaline stress response of RPY geng by modulating key genes involved in fatty acid, phenylpropanoid biosynthesis, amino acid metabolism, and glycolysis metabolic pathways based on the gene co-expression network analysis. The present research results not only provide important insights for understanding the mechanism underlying of rice saline–alkaline tolerance at the transcriptome and metabolome levels but also provide key candidate target genes for further enhancing rice saline–alkaline stress tolerance.

## 1. Introduction

Soil saline–alkalization is a widely spread abiotic stress that is emerging as a major limiting factor of global crop growth and production [1]. Soil saline–alkalization is characterized by both high salinity and high alkalinity (often pH > 8.0) due to the enhanced accumulation of two major natural alkaline salts, sodium carbonate (Na_2_CO_3_) and sodium hydrogen carbonate (NaHCO_3_) [2,3,4]. So far, extensive efforts have been conducted to dissect the underlying molecular and physiological mechanisms by which plants respond and adapt to neutral salt stress (e.g., NaCl and Na_2_SO_4_), especially in the rice model plant species (*Oryza sativa* L.) and *Arabidopsis thaliana* [1,5,6,7,8]. However, fewer studies have been devoted to the elucidation of the plants’ response and adaption to saline–alkaline stress. In contrast to neutral salt–stress–encountering problems related to osmotic stress and ionic toxicity [1,6,7,8], saline–alkaline stress is more complex, which often causes combined damages of osmotic stress, ionic toxicity, oxidative stress, and high pH stress [3,4], indicating that saline–alkaline stress will result in more severe symptoms to plants in the inhibition of growth and the antioxidant system, as well as in an imbalance of ionic homeostasis and osmotic adjustment, etc. [9,10]. For instance, a low concentration of NaHCO_3_ (<10 mM) resulted in plant leaf chlorosis and lethal symptoms, while a more high–fold concentration of NaCl (200 mM) was required to cause similar symptoms [11]. Notably, high salinity and high alkalinity often occur simultaneously in nature soils, and more than 900 million hectares of lands (>7% of the world’s total land area) are suffering from saline–alkalization all over the world. Soil saline–alkalization has become a major factor constraining the sustainable development of global agriculture. Therefore, there is an urgent need for the elucidation of the plants’ response and adaptation to saline–alkaline stress, the understanding of which can potentially provide important insights for cultivating and breeding crops capable of growth in saline–alkaline farmlands.

Saline–alkaline stress in the soil is characterized by high salt and high pH due to the hydrolysis of Na_2_CO_3_ and NaHCO_3_. High salinity is commonly known to result in a physiological water shortage on plants due to the effect of high Na^+^ concentrations in the soil, which then leads to stress pathways, including osmotic stress, ionic stress, and secondary stress, especially oxidative stress, in plants [1]. High pH in saline–alkaline soil can also reduce the uptake of many essential nutrients and the Na^+^ exclusion, thus inducing nutrient stress on plants [12]. All these result in plants suffering from osmotic stress, ionic toxicity, oxidative stress, and high pH stress simultaneously, ultimately severely destroying the cell membrane structure and its integrity, disturbing the cell pH and osmotic and ionic balances, inactivating the enzyme activity, imposing nutrient stress upon plants, etc. [3,4]. In order to cope with the adversity of saline–alkaline stress, plants have evolved multiple adaptive strategies, including mainly the accumulation of osmotic adjustment substances, such as soluble proteins and sugars, to maintain the intracellular water potential [3,4]; the excretion and isolation of Na^+^ toxicity via transporters or channel proteins or proton pumps, such as salt overly sensitives (SOSs) and high–affinity K^+^ transporters (HKTs) and H^+^–ATPase to maintain intracellular ionic homeostasis [13,14,15]; the synthesis of antioxidants (e.g., flavonoids, alkaloids, vitamins, phenolic acids, and mannitol) and antioxidant enzymes (e.g., superoxide dismutase (SOD), peroxidase (POD), and catalase (CAT)) to reduce the oxidative damage of reactive oxygen species (ROS) [3,4]; as well as the secretion of organic anions and H^+^ to maintain intracellular pH stability via acidifying the rhizosphere [15,16]. During these adaptive strategies of the plants in response to saline–alkaline stress, many genes involved mainly in ionic homeostasis, osmotic regulation, ROS-scavenging activity, as well as signal transduction are induced to express [3,17]. Previous studies have demonstrated that external saline–alkaline conditions are perceived by plants based on a series of corresponding gene changes from a complicated network of signal transduction pathways, including hormone signal transduction pathways, protein kinase pathways (e.g., mitogen–activated protein kinases (MAPKs), Ca^2+^–dependent protein kinases (CDPKs), and calcineurin–B–like interacting protein kinases (CIPKs)), and SOS pathways [17,18,19]. For instance, activated SOS2, as an intermediate hub in the SOS pathways, was responsible for the Na^+^ efflux from cytoplasm into vacuole to maintain the intracellular ion balance through phosphorylating and activating the NHX (Na^+^/H^+^ antiporter) [20,21,22]. MAPKs and receptor-like protein kinases (RLKs) played a vital role in contributing to the tolerance for plant adaption to salt stress by the modulation of ion and ROS homeostasis [19,23,24]. In addition, the concentration changes in hormones regulated by the expression of hormone-related genes were also a major factor for plant adaptation to saline–alkaline stress, such as abscisic acid (ABA), jasmonic acid (JA), brassinosteroid (BR), gibberellin (GA), and auxin (IAA) [25,26,27]. Furthermore, transcription factors (TFs), such as NACs, WRKYs, bZIPs, and bHLHs, that served as a bridge between upstream stimulus signals and their corresponding downstream–associated genes played important roles for the plants’ adaptation to saline–alkaline stress [18,28]. Moreover, the genes related to resistance, including those that are osmoprotectant–related, such as *glutathione S–transferase* (*GST*) and *glutamate decarboxylase* (*GAD*); antioxidant–related, such as *glutathione peroxidase* (*GPX*) and *pathogenesis-related proteins* (*PRs*); as well as transporter-related, such as *NHXs* and *HKTs*, were also often induced to express to regulate the intracellular balance of ions, pH, and osmotic potential in a direct way [3,4]. These findings indicate that the transcriptional regulation underlying the plants’ response and adaptation to saline–alkaline stress is a very complicated process regulated by the involvement of multiple genes in multiple metabolic pathways; nevertheless, the mechanism behind this remains largely unknown.

Rice (*O. sativa* L.), as one of the most important staple crops, is considered to be a saline–alkaline-sensitive glycophytic species [29]. In nature, high salt and high pH are often two coexisting abiotic stresses. However, in contrast to neutral salt stress, fewer studies have been conducted to understand the mechanisms of the response and adaptation to saline–alkaline stress in rice. In China, the total area of saline–alkaline soil is nearly 1 × 10^8^ hectares, accounting for approximately 10% of the total land area, which is distributed mainly in northwestern, northeastern, and coastal areas [30], and more seriously, soil saline–alkalization in these regions is continuing to expand annually at a rate of approximately 1 × 10^6^ hectares due to ecological environment deterioration, the improper application of chemical fertilizers, and poor irrigation practices [31]. Currently, approximately 20% of the total rice is planted in the saline–alkaline farmlands, and soil saline–alkalization is emerging as one of the major constraints of rice production in China [32]. Previous studies have demonstrated that different genotypes within a species show extensive variations of tolerance to saline–alkaline stress [13,33,34,35,36,37,38]. These earlier findings have focused mainly on an individual variety, identical subspecies, or their derived populations based on the analysis of physiological, hormonal, transcriptomic, or metabolomic aspects. Few research studies have been conducted to investigate the different responses to saline–alkaline stress between *indica* and *japonica* rice cultivars based on the integrated analysis of the phenotype, physiology, transcriptome, and metabolome. To our knowledge, there are marked differences in many important agronomic traits and environmental stresses (including biotic and abiotic stresses) between *indica* and *japonica* rice genotypes due to their difference in genomic structure [39,40,41,42,43,44,45,46]. It is, therefore, urgent to dissect the different underlying mechanisms by which *indica* and *japonica* rice cultivars respond and adapt to saline–alkaline stress, the understanding of which will provide the potential for cultivating the new rice cultivars with strong saline–alkaline tolerance via *indica*–*japonica* subspecies hybrids [47]. In our studies, we screened and assessed the saline–alkaline tolerance of seven rice cultivars consisting of four *indica* cultivars (93-11, Chao2R, Fuli, and Baihuamiao) and three *japonica* cultivars (Nipponbare, ZH11, and RPY geng). Among them, we found that *indica* cultivars were more sensitive to saline–alkaline stress than *japonica* cultivars. Intriguingly, hybrid F_1_ progenies from the crossing of Chao2R (*O. sativa* ssp. *indica*) and RPY geng (*O. sativa* ssp. *japonica*) showed obvious heterosis in terms of their saline–alkaline tolerance (unpublished data). The two rice genotypes, Chao2R and RPY geng, were, therefore, used to explore the underlying mechanisms by which rice responds and adapts to saline–alkaline stress at the transcriptome and metabolome levels. These findings will contribute to our understanding for breeding rice with a strong saline–alkaline tolerance.

## 2. Results

### 2.1. Chao2R and RPY Geng Showed Distinct Tolerance under Saline–Alkaline Treatment

Given that sodium carbonate (Na_2_CO_3_) and bicarbonate (NaHCO_3_) are the major alkaline salts in natural saline–alkaline environments, 0.3% (*w*/*v*) mixtures of alkaline salts (Na_2_CO_3_:NaHCO_3_ = 1:3) are added into the Yoshida solution to mimic the stress conditions that occur in saline–alkaline soils with a high pH value (9.35) and high Na^+^ concentration (approx. 40.84 mM). To compare the difference in the saline–alkaline tolerance of Chao2R (*O. sativa* ssp. *indica*) and RPY geng (*O. sativa* ssp. *japonica*), we calculated the final survival rate of both rice subspecies genotypes after exposure to saline–alkaline conditions. As shown in Figure 1, the survival rate (SR) of RPY geng (77%) was significantly higher than that of Chao2R (45%). In addition, shoot height, relative growth rate, and shoot biomass (including fresh weight and dry weight) of RPY geng seedlings were significantly higher than those of Chao2R seedlings in both the control and saline–alkaline solutions (Figure 1A and Appendix A and Table 1). Notably, shoot growth in both genotypes was observed to be reduced after the saline–alkaline treatment, and the magnitude of the reduction in shoot growth was greater in Chao2R than in RPY geng seedlings (Table 1 and Appendix A). These results demonstrate that RPY geng had a stronger tolerance to saline–alkaline stress than did Chao2R at the seedlings.

To adapt to the stress associated with the saline–alkaline soils, plants have evolved a series of strategies to cope with it, such as morphological adaption and physiological adaption (e.g., accumulation of osmoprotectants, antioxidant enzymes, and antioxidants) [3,4]. To explore the difference in the physiological response of both rice subspecies’ genotypes (Chao2R and RPY geng) under saline–alkaline conditions, we then compared the effects of saline–alkaline stress on the physiological adaption of the two genotypes. Under untreated control conditions, the proline (Pro), soluble sugar (SS), and malondialdehyde (MDA) concentrations and the total antioxidant capacity (T-AOC) of the two genotypes were comparable; the activity of superoxide dismutase (SOD) and catalase (CAT) was significantly higher in RPY geng than in Chao2R, while Chao2R had a higher peroxidase (POD) activity than did RPY geng (Table 1 and Appendix A). Compared with the untreated control conditions, the saline–alkaline stress was found to result in notable increases in Pro, SS, MDA, SOD, POD, and T-AOC in both rice seedlings, except for CAT, which had no significant differences between the two genotypes (Table 1 and Appendix A). These changes led to the magnitude of the increase in SS, POD, and T-AOC of the RPY geng seedlings being significantly greater than those of Chao2R, whereas the magnitudes of the increase in SOD, Pro, and MDA of the Chao2R seedlings were markedly greater than those of RPY geng. Among them, SS was significantly higher in RPY geng than in Chao2R, while Pro was markedly higher in Chao2R than in RPY geng, indicating that there was a significant difference between Chao2R and RPY geng in terms of the accumulation of osmoprotectants under saline–alkaline stress conditions. The production of MDA derived from lipid peroxidation can reflect the levels of cell membrane damage after exposure to environmental stresses [48]. We found the MDA concentration and its magnitude of the increase were markedly higher in Chao2R than in RPY geng, which may suggest that the degree of cell membrane damage of Chao2R was more serious than that of RPY geng under the saline–alkaline stress conditions. POD and SOD are two key antioxidant enzymes that help plants eliminate reactive oxygen species (ROS) and facilitate tolerance to environment stresses [49]. In this study, we found the activity of POD and SOD in RPY geng were significantly higher than those of Chao2R, which may indicate that RPY geng had a stronger ability to remove ROS than did Chao2R. As expected, T-AOC was markedly higher in the RPY geng than in the Chao2R seedlings with the saline–alkaline solution, which was in line with the result that RPY geng was more tolerant than Chao2R under saline–alkaline stress conditions. These results were further verified using the Pearson correlation tests between phenotypic traits and physiological characteristics: SR was significantly positively correlated with SOD and T-AOC (Appendix A)

### 2.2. Transcriptomic Analysis of Chao2R and RPY Geng in Response to Saline–Alkaline Stress

Three independent samples of both rice subspecies’ genotypes, Chao2R (*indica*, saline–alkaline-sensitive) and RPY geng (*japonica*, saline–alkaline-tolerant), were performed from the saline–alkaline treatment sample and the untreated control sample for RNA-sequencing to determine the genes responsible for the saline–alkaline stress tolerance. In total, approximately 745.07 million raw sequencing reads were generated from the BGISEQ sequencing of the 12 samples (Appendix A). Each sample covered raw reads ranging from 47.05 to 99.75 million, with an average of 62.09 million. After adapter sequences and low-quality reads were removed, the remaining clean reads of each sample ranged from 44.80 to 95.90 million, with an average of 59.41 million. Then, the 12 samples generated, in total, approximately 712.98 million clean sequencing reads and 106.94 Gb of clean bases (exceeding 6 Gb of each sample), respectively. At least 96.67% and 91.44% bases of the clean reads separately had a quality score ≥ Q20 (less than or equal to the 1% incorrect rate) and ≥ Q30 (less than or equal to the 1‰ incorrect rate), which indicated the high quality of the clean reads in each sample. Here, 92.65–96.76% of the clean reads were mapped to the rice reference genome, with an average mapping rate of 94.60% by using HISAT v2.1.0 software. There were 30,950, 29,344, 30,393, and 30,054 predicted genes of the clean reads mapped in the rice reference genome in Chao2R (Chao2R with untreated control conditions), Chao2RA (Chao2R treated with saline–alkaline conditions for 72 h), RPY (RPY geng with untreated control conditions), and RPYA (RPY geng treated with saline–alkaline conditions for 72 h), respectively. Therefore, the results indicated that more than half of the rice reference genes were expressed in each sample, and these genes were sufficient for the following differential gene expression analysis: The global gene expression levels were similar among the three biological replicates of each treatment (Figure 2A), which suggested good repeatability in each treatment. Principal component analysis (PCA) was performed to evaluate the overall differences of all detected genes between the two rice genotypes under the different treatments. Chao2R and RPY geng could be clearly divided into different regions based on the first principal component (PC1) and PC2, which highlighted the great influence of saline–alkaline stress on the change in the detected genes in the two rice genotypes (Figure 2B). PC1 (35.05%) and PC2 (23.87%) together explained 58.92% of the total variation. In addition, biological replicates from the same treatment were obviously clustered together, which further indicated good repeatability in each treatment.

Differential expression genes (DEGs) in this study were designated as those genes that showed at least a twofold change (the absolute value of the log_2_ (fold change with FPKM) *≥* 1) and were determined to be significantly different (false discovery rate (FDR) < 0.05) under saline–alkaline stress compared with the control. RPY geng (10,117) had more DEGs than did Chao2R (8,623), and more genes were downregulated than upregulated in both rice genotypes (Figure 2C). It was also apparent that RPY geng had more specific DEGs than did Chao2R in either the downregulated or the upregulated fraction (Figure 2D), revealing an apparent difference between the two rice subspecies in response to the saline–alkaline stress at the transcriptional level. Combined with the data of the survival rates (Figure 1B), this could reasonably suggest that RPY geng was more tolerant to saline–alkaline stress than was Chao2R due to its more rapid response to the saline–alkaline stress at the level of transcription.

To further determine the saline–alkaline responsive pathways of the two rice genotypes, both the Gene Ontology (GO) and Kyoto Encyclopedia of Genes and Genomes (KEGG) approaches were used to investigate the functional enrichment and annotation analysis of the DEGs. Both approaches showed that Chao2R and RPY geng had modified their metabolism pathways after exposure to the saline–alkaline treatment. In the GO analysis (Figure 3 and Appendix A), the GO category biology process ‘response to salt stress’ and ‘response to osmotic stress’ were enriched in the upregulated DEGs of both Chao2R and RPY geng. The lipid metabolic pathway was especially enriched in the upregulated DEGs of RPY geng, including the biology process categories of ‘fatty acid catabolic process’, ‘fatty acid β-oxidation’, ‘fatty acid oxidation’, ‘lipid oxidation’, ‘cellular lipid catabolic process’ and ‘lipid modification’. However, the GO terms of the upregulated DEGs of Chao2R were associated mainly with the chitin-related process and water-responsive processes, including the categories of ‘response to chitin’, ‘chitinase activity’, ‘chitin binding’, ‘chitin metabolic process’, ‘chitin catabolic process’, ‘response to water’, ‘response to water deprivation’, and ‘positive regulation of response to water deprivation’. The findings implied a different saline–alkaline responsive expression mode of the two rice subspecies’ genotypes. In contrast, the downregulated DEGs of both Chao2R and RPY geng were enriched mainly in ‘cell structure’, ‘cell cycle process’, and ‘photosynthesis process’. It could be inferred that both Chao2R and RPY geng may have transferred the energy requirements of the cell structure, the cell cycle process, and the photosynthesis process to produce greater metabolites, such as lipids and chitins. In the KEGG analysis, similar conclusions could be drawn (Appendix A). Several pathways were enriched in the upregulated DEGs of both Chao2R and RPY geng, including the categories of ‘α-linolenic acid metabolism’, ‘butanoate metabolism’, ‘propanoate metabolism’, ‘galactose metabolism’, ‘valine, leucine, and isoleucine degradation’, ‘β-alanine metabolism’, as well as ‘arginine and proline metabolism’. Notably, the plant signal transduction pathway was also enriched in the upregulation of both Chao2R and RPY geng, with the KEGG pathway ‘plant hormone signal transduction’ upregulated in RPY geng and ‘MAPK signaling pathway-plant’ upregulated in Chao2R. The lipid and amino acid metabolic pathways were specifically enriched in the upregulated DEGs of RPY geng, including the KEGG pathways of ‘fatty acid degradation’, ‘glycerolipid metabolism’, ‘biosynthesis of unsaturated fatty acids’, ’phenylalanine metabolism’, ‘tyrosin metabolism’, and ‘tryptophan metabolism’. In contrast, many pathways were enriched in the downregulated DEGs of both Chao2R and RPY geng genotypes, such as ‘ribosome’, ‘photosynthesis’, ‘aminoacyl-tRNA biosynthesis’, ‘photosynthesis-antenna proteins’, and ‘glycine, serine, and threonine metabolism’. In all, the findings of the functional enrichment and annotation of the DEGs indicated that Chao2R and RPY geng modified the expression of multiple genes involved in many pathways, including lipid metabolic, amino acid and organic acid metabolic, terpenoid biosynthesis, plant signal transduction, chitin-related, as well as salt and osmotic stress responsive, which may suggest their involvement in rice saline–alkaline stress resistance. The genes related to lipid metabolism, phenylalanine metabolism, tryptophan metabolism, tyrosine metabolism, signal transduction, and sugar metabolism that were especially enhanced by the saline–alkaline stress in RPY geng may play exceptional roles in building up its resistance against saline–alkaline stress.

### 2.3. Metabolomic Analysis of Chao2R and RPY Geng in Response to Saline–Alkaline Stress

To determine how metabolites respond to saline–alkaline stress, the 12 samples taken from both rice genotypes (Chao2R and RPY geng) during saline–alkaline treatment and untreated control were next subjected to wide-targeted metabolic profiling using UPLC-ESI-MS/MS. Here, a total of 801 metabolites were identified and annotated among the 12 samples, which could be divided into 13 different categories, including 74 amino acids and their derivatives (9.24%), 131 lipids (16.35%), 49 nucleotides and their derivatives (6.12%), 78 organic acids (9.74%), 42 saccharides and alcohols (5.24%), 18 vitamins (2.25%), 73 alkaloids (9.11%), 189 flavonoids (23.60%), 18 types of lignans and coumarins (2.25%), 108 phenolic acids (13.48%), 7 tannins (0.87%), 6 terpenoids (0.74%), and 8 types of other compounds (1.00%) (Figure 4A). The overall metabolic profile of Chao2R and RPY geng in each treatment could be visualized based on hierarchical cluster analysis (HCA) (Figure 4B). The heat map showed that the three biological replicates from the same group were clustered together, which suggested a high repeatability and a robust correlation between replicates. Flavonoids, lipids, phenolic acids, organic acids, amino acids and their derivatives, and alkaloids were six dominant metabolites based on the type and proportion of the metabolites. PCA was subsequently used to evaluate the overall differences between Chao2R and RPY geng under the different treatments at the metabolic level. Chao2R and RPY geng could be obviously separated into distinct regions by PC1 and PC2, indicating that the saline–alkaline conditions could have a great impact on the change of the metabolites in both rice subspecies’ genotypes (Figure 4C). The PCA results of all identified metabolites showed that the first two PCs explained 64.98% of the total variation. Similar to gene expression (Figure 2B), all the biological replicates of metabolomics samples were also grouped together, indicating the high repeatability and good homogeneity among the replicates.

Both rice subspecies’ genotypes showed a distinctive saline–alkaline responsive accumulation pattern at the metabolic level. Here, significantly differentially accumulated metabolites (DAMs) between the groups in this study were defined as those with at least a twofold alteration (|log_2_FC (fold change)| *≥* 1) and a variable importance in projection (VIP) *≥* 1. In all, 560 DAMs were identified between Chao2R and RPY geng under the saline–alkaline treatment in comparison with the control condition, including mainly flavonoids (24.46%), lipids (16.96%), phenolic acids (13.21%), alkaloids (10.36%), organic acids (10.36%), and amino acids and their derivatives (9.46%) (Appendix A). More DAMs were observed in RPY geng (446) than in Chao2R (392); more metabolites were downregulated than upregulated in Chao2R, while more metabolites were upregulated than downregulated in RPY geng (Figure 4D and Appendix A). There were 278 DAMs common in Chao2R and RPY geng, with the DAMs specifically accumulated in both rice subspecies’ genotypes: 114 in Chao2R and 168 in RPY geng (Appendix A). In particular, RPY geng had more unique DAMs than did Chao2R in the upregulated fraction, while more specific DAMs were detected in Chao2R than RPY geng in the downregulated fraction (Figure 4E).

The DAMs identified in both rice genotypes were further investigated using the KEGG pathway database. The KEGG analysis revealed that Chao2R more so than RPY geng had modified their metabolism after exposure to the saline–alkaline conditions. Notably, completely different pathways were specifically enriched in the DAMs between Chao2R and RPY geng, respectively (Appendix A). Five pathways were significantly enriched in the 169 DAMs of RPY geng, including ‘tryptophan metabolism’, ‘glucosinolate biosynthesis’, ‘aminoacyl-tRNA biosynthesis’, ‘phenylalanine metabolism’, and ‘tropane, piperidine, and pyridine alkaloid biosynthesis’, while the significant categories among 9 pathways in the 137 DAMs of Chao2R included ‘pyruvate metabolism’, ‘valine, leucine, and isoleucine degradation’, ‘2-oxocarboxylic acid metabolism’, ‘citrate cycle (TCA cycle)’, ‘propanoate metabolism’, ‘carbon metabolism‘, ‘biosynthesis of secondary metabolites ‘, ‘valine, leucine, and isoleucine biosynthesis’, and ‘biosynthesis of amino acids’. To observe more intuitively the variations occurring among these different metabolites in Chao2R and RPY geng during the saline–alkaline conditions and the untreated control, a heat map of the 560 different metabolites was obtained (Appendix A). Among them, 108 metabolites increased significantly in both Chao2R and RPY geng after exposure to the saline–alkaline treatment when compared with their levels in the untreated control (Figure 4E and Appendix A), including 25 alkaloids, 23 amino acids and their derivatives, 21 lipids, and 18 organic acids, indicating these metabolites were most involved in the rice’s saline–alkaline tolerance. The 229 metabolites that induced specifically in RPY geng consisted of lipids (56), flavonoids (48), phenolic acids (40), and alkaloids (21), etc., and this could be an indication of their exceptional functional involvement in the saline–alkaline tolerance of RPY geng. In contrast, those 86 metabolites that increased uniquely in Chao2R were flavonoids (39), organic acids (13), phenolic acids (10), etc. Subsequently, 560 DAMs were classified, and the changes in the various types of metabolites were observed between Chao2R and RPY geng after exposure to the saline–alkaline conditions in comparison with the control. The results showed that many types of metabolites displayed distinct changes and trends in both rice genotypes, such as lignans and coumarins, nucleotides and their derivatives, vitamins, lipids, alkaloids, as well as phenolic acids (Figure 4F and Appendix A). It is noteworthy that the content and alteration of lipids, alkaloids, and phenolic acids in RPY geng were significantly higher than those in Chao2R. In contrast, higher contents of organic acids, tannins, terpenoids, as well as saccharides and alcohols were detected in Chao2R than in RPY geng. Based on the above results, a clear difference was observed between the two rice subspecies’ genotypes in response to the saline–alkaline stress at the metabolic level. Together with the survival data (Figure 1B), it could be speculated that RPY geng was more tolerant to the saline–alkaline stress than Chao2R by rapidly and specifically increasing the metabolites under the saline–alkaline stress treatment, especially lipids, alkaloids, as well as phenolic acids.

### 2.4. Mining the Potential Important Candidate Genes and Metabolites for Saline–Alkaline Tolerance

In order to explore further the molecular mechanisms of RPY geng in regulating and adapting to salt–alkaline stress more than Chao2R, an integrated systemic KEGG mapping analysis was constructed based on the differences in the metabolites and genes. A total of 77 metabolic pathways displayed alterations in the rice in response to the saline–alkaline stress, and 1504 genes and 169 metabolites were mapped to these pathways (Appendix A). As shown in Figure 5, these metabolites and genes were involved mainly in the pathways related to amino acid metabolism, energy metabolism (Glycolysis and TCA cycle), fatty acid metabolism, alkaloid biosynthesis, phenylpropanoid biosynthesis, as well as plant signal transduction pathway (hormone signal transduction and MAPK signaling pathways).

Considering all the results of the transcriptome and metabolome, we then focused on DEGs and DAMs which showed the opposite regulation by the saline–alkaline condition in RPY geng and Chao2R to narrow down the number of potential important candidate genes and metabolites that were specifically regulated by the saline–alkaline stress. We speculated that these genes and metabolites may play different roles in the RPY geng and Chao2R responses to the saline–alkaline stress. Based on the criteria, we finally screened 68 genes and 53 metabolites (including 19 lipids, 14 phenolic acids, 8 alkaloids, 4 lignans, 3 vitamins, 3 amino acid, and 1 coumarin), those most involved in the rice saline–alkaline tolerance (Figure 6A,B). Amongst these genes and metabolites, most were specifically upregulated by the saline–alkaline stress in RPY geng. The functional correlation of these 68 genes in response to the saline–alkaline stress was firstly tested by constructing a co-expression network using Cytoscape v3.8.2 software (Pearson correlation coefficient (PCC) ≥ 0.9, *p* < 0.01). Based on gene connectivity, it was found that some genes involved in the plant signal transduction pathway were identified as hub genes, such as *PR5* (*pathogenesis-related protein*, *Os02g12510*), *WRKY* (*Os08g0235800* and *Os07g0111400*), *SPCH*/*bHLH* (*Os02g0691500*), *NAC* (*Os02g0285900*), *FLS2* (*Os04g0349700*), *LUG* (*Os02g0813800*), *BRI1* (*protein brassinosteroid-insensitive 1*, *Os02g0154200*), as well as *TGA*/*bZIP* (*Os05g0443900*), and these hub genes were found to be highly co-expressed with genes involved mainly in the amino sugar and nucleotide sugar metabolism (especially *Os08g0237200* and *Os03g0268400*), glutathione metabolism (*Os06g0185900* and *Os10g0525500*), phenylpropanoid biosynthesis (especially *Os08g0543400*, *Os04g0473900*, and *Os01g0591300*), ubiquitin-mediated proteolysis (especially *Os05g0182500*, *Os09g0272900*, and *Os05g0193900*), fatty acid metabolism (especially *Os03g0108500*, *Os06g0300000*, *Os08g0184800*, *Os12g0559200*, and *Os12g0102100*), amino acid metabolism (especially *Os04g0107500* and *Os04g0447800*), glycolysis (especially *Os10g0204400* and *Os07g0197100*), etc. (Figure 6C). Therefore, we may speculate that these genes related to the plant signal transduction pathway play important roles in regulating the key genes’ involvement in sugar metabolism, amino acid metabolism, energy metabolism, fatty acid metabolism, phenylpropanoid biosynthesis, etc., and a series of these genes’ cascade correlations may accelerate and drive the higher saline–alkaline tolerance of RPY geng compared with that of Chao2R.

In order to validate the reliability of the transcriptome datasets under study, twenty potential candidate genes related to the plant signal transduction pathway, sugar metabolism, glutathione metabolism, phenylpropanoid biosynthesis, fatty acid metabolism, amino acid metabolism, as well as glycolysis were selected for study by quantitative reverse transcription-polymerase chain reaction (qRT-PCR), with gene-specific primers in all samples of Chao2R and RPY geng (Appendix A). All twenty genes showed a largely consistent expression pattern between the qRT-PCR results and the RNA-seq data (R^2^ = 0.9573, *p* < 0.0001) (Appendix A), revealing that the RNA-seq dataset under study was highly adequate for further analysis.

To reveal the correlation between the 68 genes and 53 metabolites involved in the rice saline–alkaline stress tolerance, an interactive network between these genes and the metabolites was constructed using Cytoscape v3.8.2 software based on the screening criteria of the absolute value of PCC ≥ 0.9 and *p* < 0.01. As shown in Figure 6D, there are 52 metabolites (except for N-Acetylornithine) correlated with 67 genes (except for *Os01g0952800*). For instance, 5 genes (especially *Os08g0543400*, *Os04g0473900*, and *Os01g0591300*) related to phenylpropanoid biosynthesis were positively correlated with 48 metabolic compounds, including 18 lipids, 13 phenolic acids, 8 alkaloids, 4 lignans, 3 vitamins, 1 coumarin, and 1 amino acid; there were 12 genes (especially *Os04g0146800*, *Os06g0300000*, *Os12g0559200*, and *Os03g0423300*) involved in fatty acid metabolism that were positively related with 50 metabolites, including 19 lipids, 13 phenolic acids, 9 alkaloids, 4 lignans, 3 vitamins, 1 coumarin, and 1 amino acid; another 5 genes (especially *Os08g0237200* and *Os11g0204600*) related to amino sugar and nucleotide sugar metabolism were positively correlated with 46 metabolic compounds; there were 6 genes (especially *Os10g0213100*, *Os10g0155400*, and *Os04g0447800*) mapped to amino acid metabolism that were positively related with 40 metabolites; another 5 genes (especially *Os06g0145600* and *Os02g0734400*) assigned to cutin, suberine, and wax biosynthesis were positively correlated with 35 metabolic compounds; we found 3 genes (especially *Os07g0197100* and *Os10g0204400*) involved in glycolysis that were positively related with 32 metabolites; there were 3 genes (especially *Os05g0182500* and *Os09g0272900*) related to ubiquitin-mediated proteolysis that were positively correlated with 42 metabolites; lastly, 2 genes (*Os10g0525500* and *Os06g0185900*) involved in glutathione metabolism were positively related with 30 metabolites. Notably, 19 genes assigned to the plant signal transduction pathway were also positively correlated with 50 metabolic compounds, including 19 lipids, 13 phenolic acids, 9 alkaloids, 4 lignans, 3 vitamins, 1 coumarin, and 1 amino acid, indicating that the involvement of these genes (especially *Os02g12510*, *Os04g0349700*, *Os06g0306600*, *Os08g0235800*, *Os02g0154200*, *Os05g0443900*, *Os07g0111400*, and *Os12g0609200*) in the plant signal transduction pathway may have played important roles in positively regulating the biosynthesis of these 50 metabolites.

## 3. Discussion

Rice (*O. sativa* L.), as one of the most important staple crops, is a saline–alkaline-sensitive crop. Currently, approximately 20% of the total rice is planted in saline–alkaline soil, and soil saline–alkalization has become one of the major constraints of rice production [32]. In contrast to neutral salinity, fewer related studies have been conducted on saline–alkaline stress. However, high salt concentration and high pH are often two coexisting abiotic stresses in nature. Therefore, there is an urgent need to dissect the mechanisms regarding how rice regulates and adapts to this saline–alkaline stress, the understanding of which can potentially increase the rice saline–alkaline tolerance and exploit adequately the saline–alkaline farmlands. Sodium carbonate (Na_2_CO_3_) and bicarbonate (NaHCO_3_) are the two major salts existing in natural saline–alkaline farmlands, which can cause the synergistic damages of high Na^+^ toxicity, osmotic stress, and pH stress on rice [2,3,4]. Here, we have shown that RPY geng (*japonica*) is a more saline–alkaline-tolerant genotype than Chao2R (*indica*), as evidenced by a lower magnitude of reductions in shoot height and shoot biomass, lower magnitude of increase in MDA, as well as more marked increases in SR, SS, POD, SOD, and T-AOC (Figure 1 and Appendix A and Table 1), indicating that an in-depth investigation of the different mechanisms between *indica* Chao2R and *japonica* RPY geng responses to saline–alkali stress might provide a new insight for improving rice saline–alkali tolerance.

Given the deleterious effects of saline–alkaline stress on rice, previous some studies have been conducted to investigate the potential mechanisms by which rice regulates and adapts to saline–alkaline stress conditions. For instance, *japonica* ‘Nipponbare’ was used to alleviate the harmful effect of the saline–alkaline stress using magnetized water [50]; *japonica* ‘Tongxi926′ was used to dissect the effects of long-term high saline–alkaline stress on rice [36]; *indica* ’93-11′ and its polyploidy were used to explore the mechanisms of saline–alkali tolerance [51]; two *japonica* cultivars ‘dongdao-4′ and ‘jigeng-88′ were used to study their differences in tolerance to saline–alkaline stress [16,35,37]; and two *indica* cultivars ‘FL478′ and ‘IR29′ were used to reveal their differences under saline–alkaline stress by regulating Na^+^ transport [13]. These earlier studies have focused mainly on an individual variety, identical subspecies, or their derived populations based on the analysis of physiological, hormonal, transcriptomic, or metabolomic aspects. In contrast, fewer studies have been conducted to explore the different responses to saline–alkaline stress between *indica* and *japonica* rice cultivars based on the comprehensive phenotypic, physiological, transcriptomic, and metabolomics analyses. As we know, there are marked differences in many important agronomic traits and environmental stresses (including biotic and abiotic stresses) between *indica* and *japonica* rice cultivars due to the difference in their genomic structures [39,40,41,42,43,44,45,46]. In order to take full advantage of the differences between *indica* Chao2R (saline–alkaline-sensitive genotype) and *japonica* RPY geng (saline–alkaline-tolerant genotype) for breeding new rice cultivars with higher saline–alkaline stress tolerance via the *indica*–*japonica* subspecific hybrids, there is first an urgent need to explore the different mechanisms by which they regulate and adapt to saline–alkaline stress at the transcriptional and metabolomic levels. In our study, the lipid metabolic pathway, the plant signal transduction pathway, as well as the secondary metabolic biosynthesis (including phenolic acid and alkaloid) that were specifically enriched by the saline–alkaline stress conditions in RPY geng may play important roles in building up its resistance against saline–alkaline stress according to the GO and KEGG aspects (Figure 3, Figure 5 and Appendix A), suggesting that RPY geng was equipped with a more efficient mechanism by coupling with multiple biological processes for adapting to the saline–alkaline conditions. Recently, QTL mapping and candidate gene analysis for rice saline–alkaline tolerance has been explored based on the genome-wide association study (GWAS) and linkage-mapping approaches using bi-parental populations (such as recombinant inbred lines (RILs)) or natural populations [29,33,38]. For example, linkage mapping and a GWAS analysis had been used to detect 3 candidate genes (*Os11g37300*, *Os11g37320*, and *Os11g37390*) within a major QTL (*qAT11*) on chromosome 11 for alkali tolerance in *Japonica* rice using 184 RILs and 295 *japonica* rice varieties [38]. Similarly, one candidate gene (*OsHKT1;1*) within a major QTL (*RNC4*) significantly associated with root Na^+^ content and root Na^+^/K^+^ ratio was identified for regulating the distinct salt tolerance between *indica* accessions and *japonica* accessions by GWAS [52]. Li et al. (2019) used 295 *japonica* rice varieties to evaluate the alkalinity tolerance at the seedling stage via GWAS, and *Os03g26210* (*OsIRO3*) within a common QTL on chromosome 3 associated with the three phenotypes of alkalinity tolerance was identified for improving the alkalinity tolerance in rice [33]. These results suggest that QTL mapping and GWAS in bi-parental populations and natural populations are powerful strategies for dissecting the genetic variations and identifying the candidate genes in saline–alkaline tolerance among the rice germplasms. Therefore, major QTLs or candidate genes related to the saline–alkaline-tolerant differences between *indica* ‘Chao2R’ and *japonica* ‘RPY geng’ will be expected to be more finely mapped using the comprehensive analysis of the phenotypical, metabolomics, transcriptomic, and genomic datasets based on the linkage-mapping populations consisting of 270 derived RILs from a cross between Chao2R and RPY geng in our lab [39]. Notably, QTL mapping in bi-parental populations is insufficient to reveal the genetic architecture and identify the loci and candidate genes in rice saline–alkaline tolerance due to the difficulty in the acquisition of effective phenotypic data [29,53]. Having effective phenotypic data is a prerequisite for the detection of QTLs or genes for the saline–alkaline-tolerance trait. In contrast to the QTL linkage analysis in bi-parental segregating populations, GWAS is a more powerful tool for gaining insight into the valuable natural variations in trait-associated loci or genes, thus detecting multiple alleles at the same site based on high-density variations in the natural populations [33]. In order to improve the accuracy and breadth of major QTL loci, it is expected that mining the candidate genes or QTLs significantly associated with saline–alkaline tolerance in *japonica* ‘RPY geng’ will be carried out in the future through the combination analysis of bi-parental QTL mapping and GWAS.

Saline–alkaline tolerance is a very complicated quantitative trait controlled by multiple genes in the rice [29]. External saline–alkaline stress can induce expression changes of a series of cascade-related genes that are involved mainly in ionic homeostasis, osmotic regulation, ROS-scavenging activity, as well as signal transduction [3,4]. Herein, we ultimately screened the 68 genes that are most involved in the rice saline–alkaline tolerance based on an integrated analysis of transcriptome and metabolome (Figure 6A). Notably, the largest portion of these genes was most enriched in the plant signal transduction pathways. Previous studies have demonstrated that external saline–alkaline conditions were firstly perceived by plants based on a complicated network of signal transduction pathways, including hormone signal transduction pathways (e.g., ABA and IAA), protein kinase pathways (e.g., MAPKs, CDPKs, and CIPKs), and salt overly sensitive (SOSs) pathways [6,7,8,17]. The SOS pathway, as the first signal transduction pathway established in plants during saline–alkaline stress, was responsible for the Na^+^ efflux in the cells [14]. The activated SOS2 that acted as an intermediate hub in the SOS pathways could activate the NHX (Na^+^/H^+^ antiporter) by phosphorylation to maintain the intracellular ion balance through the efflux of Na^+^ from the cytoplasm into the vacuole [20,21,22]. *NHX*-encoding reverse transport protein/channel ion could enhance the plant salt tolerance through the modulation of a high Na^+^ efflux rate and K^+^/Na^+^ ratio by inducing the expression of *SKOR* (*stellar K^+^ outward rectifier*), *SOS1* (*salt overly sensitive 1*), and *AKT1* (*Arabidopsis K^+^ transporter 1*) [54,55,56]. Plants could also respond to the saline–alkaline stress through the regulation of protein kinases consisting of mitogen-activated protein kinases (MAPKs) [19,23,57] and receptor-like protein kinases (RLKs) [24,58], and overexpressing these family members in transgenic plants significantly enhanced the tolerance for plant adaption to salt stress by modulating the ion and ROS homeostasis. In addition, alterations in the hormones are an indispensable factor for plant adaption to growth and development during signal transduction pathway involvement in response to the saline–alkaline stress [25,26,27]. For instance, the involvement of *BRI1* (*brassinosteroid*-*insensitive 1*) in the brassinosteroid (BR) signaling pathway was induced to express by salinity or alkalinity stress, and its overexpression in transgenic plants was more tolerant to salinity or alkalinity [59,60]; *GID1* (*gibberellin-insensitive dwarf 1*) that could function as a soluble GA receptor [61], was found to be involved in the response to salt or alkali stresses by modulating the endogenous GA levels [37]; ABCGs that functioned as an ABA exporter or importer participated in the plants’ adaption to salt stress by cooperating to transport ABA and Na^+^ in this process [62]; Aux/IAAs, as one of primary protein components associated with the auxin signaling pathway, mediated the distribution and accumulation of IAA in plant roots in response to salt or alkali stresses [63,64]. Studies have also demonstrated that the transcript levels of genes involved in JA biosynthesis and its signal pathway were significantly upregulated in the plant’s response to high salinity or alkalinity stress, such as *AOC* (*allene oxide cyclase*), *AOS* (*allene oxide synthase*), *LOX* (*lipoxygenase*), *MFP* (*multifunctional protein*), *OPR* (*12-oxophytodienoate reductase*), and *JAZ* (*jasmonate ZIM-domain protein*), and their overexpressions in transgenic plants were more tolerant to salinity or alkalinity [65,66,67,68]. In this study, genes encoding the protein homologs of NHX (Os06g0318500), SKOR (Os01g0189100), MKK (Os02g0694900), RLK (Os01g0137700), BRI1s (Os02g0154200, Os11g0618300, Os06g0544100, and Os08g0521200), GID1s (Os08g0475400 and Os06g0306600), ABCG11 (Os12g0411700), Aux/IAA (Os03g0797800), as well as JAZ (Os12g0609200) were rapidly and specifically induced by saline–alkaline stress in RPY geng, suggesting that the plant signal transduction pathways may play vital roles in rice for saline–alkaline stress tolerance.

Transcription factors act as a bridge between upstream stimulus signals and a series of their corresponding downstream–associated genes along the signal transduction pathway in response to the saline–alkaline stress [18,28]. For example, *NACs* encoding the plant-specific transcription factor family were reported to be highly linked to the responses to salinity and alkalinity stress through the adjustment of the intracellular ion homeostasis, osmotic potential, and the ROS-scavenging capability [69,70], which may participate in the ABA- or ethylene-dependent signal transduction pathway [1,71,72]. *WRKY* was induced to express during salinity or alkalinity, and the overexpression of *WRKY* in transgenic plants contributed to the salt or alkali tolerance by enhancing the contents of the osmotic adjustment compounds (such as proline and soluble proteins) and reducing the ROS levels [73,74]. Recent studies have shown that salt or alkali stress could trigger the transcript levels of *bZIP* transcription factors, and *bZIP*-overexpressing plants increased the salinity or alkalinity resistance by maintaining an ion homeostasis, contributing to the ROS detoxification and reducing the MDA accumulation based on an ABA-dependent signal transduction pathway [75,76]. In addition, a study has proven that the *bHLH* transcription factor functioned positively in salt tolerance through its effects on the intracellular ion homeostasis and proline biosynthesis by directly activating the expression of the *SOS1* and *P5CS* (*pyrroline-5-carboxylate acid synthetase*) gene homologs [77]. In this study, several transcription factors, such as *NAC* (*Os02g0285900*), *WRKYs* (*Os07g0111400* and *Os08g0235800*), *bZIP* (*Os05g0443900*), as well as *bHLHs* (*Os01g0952800* and *Os02g0691500*), were significantly and uniquely upregulated in the RPY geng response to saline–alkaline stress, suggesting these transcription factors could be involved in the saline–alkaline tolerance of rice by mediating their corresponding related pathways.

Given the adversity associated with saline–alkaline stress, the genes related mainly to ionic homeostasis, osmotic regulation, and ROS-scavenging activity were specifically induced to maintain an intracellular balance of ions, pH, and osmotic potential in a direct way [3,4]. Herein, the transcriptome analysis showed that some candidate genes were specifically upregulated in RPY geng under saline–alkaline stress, including three-, five-, six-, five-, twelve-, five-, two-, and three-gene homologs related to glycolysis/gluconeogenesis metabolism (*ATP*, *ADH*, and *HK*), amino sugar and nucleotide sugar metabolism (e.g., *MPGs*), biosynthesis of amino acids (e.g., *GAD*), phenylpropanoid biosynthesis (e.g., *PAL* and *4CL*), fatty acid metabolism (e.g., *PLA2*, *ACS*, *MFP2*, *OPR*, *LOX*, and *ACX*), cutin, suberine, and wax biosynthesis (e.g., *POGs*, *WAX2*, and *CER1*), glutathione metabolism (*GST* and *GPX*), as well as ubiquitin-mediated proteolysis (e.g., *UBC* and *SKP1*), respectively (Figure 6A). Previous studies revealed that these genes played particular roles in the response and adaptation to the saline or alkaline stress conditions. For instance, glucose, as a signal, regulated the plant growth, development and stress response, and hexokinases (HKs) functioned in the sugar-mediated sensing and signaling pathways to regulate and adapt to the salinity tolerance by interacting with and phosphorylating NHX1 [78,79,80]. *Alcohol dehydrogenase* (*ADH*) could be induced to express in the plants’ response to the salinity stress, and the *ADH*-overexpressing plants could contribute to the interconversion of aldehydes to alcohols; then, this occurrence might trigger the production of green leaf volatiles (GLVs), thereby increasing the expression levels of multiple stress-related genes and the accumulation of callose depositions and soluble sugars, ultimately conferring enhanced resistance to the salinity stress of the plants [81,82]. ATP, as one of the key salt-signaling network molecules, played important roles in the plant adaption to salinity stress through triggering ethylene to maintain K^+^/Na^+^ homeostasis by activating and strengthening the H_2_O_2_ and Ca^2+^ signaling pathways [83,84]. Recent new candidate genes encoding mannose-1-phosphate guanyl transferases (MPGs) were identified by the functional screening of a cDNA library from a salt-tolerant rice, and the overexpression of *MPG1* in the transgenic lines played a particular role in enhancing the salinity tolerance [85]. In addition, glutamate decarboxylase (GAD) was involved in the γ-aminobutyric acid (GABA) biosynthesis, and the increasing endogenous GABA levels alleviated the plants’ effects of salt damage through modulating the Na^+^ uptake, increasing the amino acid level, as well as strengthening the antioxidant metabolism to decrease the ROS and MDA contents [86,87]. In addition, several key genes (e.g., *PALs* and *4CLs*) involved in the phenylpropanoid biosynthesis may have contributed to the salinity or alkalinity tolerance by triggering the ROS-scavenging activity according to recent comparative omics analyses [88,89]. Moreover, fatty acids and their derivatives participated in the plant tolerance to stress response together with their roles in lipid-dependent signaling cascades [90,91]. The involvement of acyl-coenzyme A oxidases (ACXs) in the β-oxidation played an important role in the plant growth, development, and stress response, and the overexpression of the *ACX* gene in the plants enhanced their drought and salinity tolerance by reducing the experienced oxidative stress [92]. Long-chain acyl-CoA synthetases (ACSs) participated in plant wax biosynthesis, and overexpressing *ACS* transgenic plants were more tolerant to salt stress through their accumulation of cuticular wax together with reduced water loss rates and epidermal permeability [93,94]. Similarly, *Eceriferum1* (*CER1*) and *Eceriferum3* (*WAX2*) also played pivotal roles in cuticular wax biosynthesis, and the transgenic lines of *CER1*- and *WAX2*-overexpression affected the plants’ response to the biotic and abiotic stresses by altering the composition and ultrastructure of the cuticular waxes [95,96]. Furthermore, given that saline–alkaline stress often results in the generation and accumulation of ROS, the plants had evolved a set of scavenging systems to maintain the intracellular ROS balance consisting of antioxidant enzymes, such as peroxidase (POD), catalase (CAT), glutathione peroxidase (GPX), and glutathione S-transferase (GST) [3,4]. *GST*-overexpressing plants induced the accumulation of proline and soluble sugar together with the expression of three ROS detoxification-related genes (CAT, POD, and SOD), thereby enhancing the activity of CAT, POD, and SOD, which, in turn, reduced the content of ROS and MDA, ultimately promoting the resistance to saline–alkaline stress [97]. Ubiquitination played vital roles in the plant stress responses, and the overexpressed *UBC* gene encoding a ubiquitin-conjugating enzyme contributed to the salt and drought stress tolerance by adjusting the transcription level of the genes related to the oxidative stress responsive, osmolyte synthesis, and ion homeostasis in the plants [98]. Similarly, S-phase kinase-associated protein 1 (SKP1), as one of the main important components of the SKP1/Cullin/F-box (SCF) complex, was involved in multiple functions, including control of the cell cycle progression, signal transduction, and transcriptional regulation. The overexpression of *SKP1* conferred salinity tolerance by enhancing the expression of the genes’ involvement in plant reproduction, growth, and stress response [99]. Additionally, *Os02g12510* and *Os03g0160600* were identified as novel gene homologs of *PR5* (*pathogenesis-related protein 5*) and *5PTase4* (*phosphatidylinositol 5-phosphatases 4*) under study, respectively. The PRs were regarded as a group of heterogeneous proteins encoded by the genes that were rapidly regulated by environmental factors consisting of pathogenic infections, abiotic stresses, and hormones (e.g., SA, JA, ABA, brassinosteroid, and ethylene), suggesting that they played important roles in protecting the plants against certain stress responses (e.g., pathogens, salt, and drought) and developmental processes [100,101]. 5PTases were key components of the intracellular vesicle-trafficking system involved in plant salt tolerance through the coordination of the ROS production, endocytosis, calcium influx, and the induction of stress-responsive genes [102,103]. In all, from these transcriptome findings under study, it can be speculated that the molecular mechanism of saline–alkaline tolerance in rice is a very complicated processes controlled by multiple genes’ involvement in the plant signal transduction, sugar and fatty acid metabolism, amino acid and secondary metabolite biosynthesis, ubiquitin-mediated proteolysis, etc., suggesting that the mechanism behind the saline–alkaline resistance in rice may be the result of a series of cascade regulations of the genes’ involvement in multiple metabolic pathways. In particular, the transcription factors *NAC* (*Os02g0285900*) and *WRKYs* (*Os07g0111400* and *Os08g0235800*) were highly co-expressed with the genes related to plant signal transduction (*PR5* (*Os02g12510*), *BRI1* (*Os02g0154200*), and *FLS2* (*Os04g0349700*)), secondary metabolite biosynthesis (HCT (Os08g0543400) and 4CL (Os04g0473900)), glutathione metabolism (GPX (Os06g0185900)), ubiquitin-mediated proteolysis (UBC (Os05g0182500) and Os09g0272900 (RPS2)), fatty acid metabolism (Os03g0108500 (SMO1-1)), biosynthesis of amino acids (Os04g0107500 (HPR)), as well as sugar metabolism (Os08g0237200 (MPG)), indicating that the three transcription factors may have potential functions in regulating these metabolism-related genes in response to saline–alkaline stress (Figure 6C). However, it is worth further investigating this issue to obtain a full understanding of the mechanism behind these gene regulations in response to saline–alkaline tolerance.

The metabolome analysis results in this study showed that more specific DAMs were observed in RPY geng than in Chao2R in the upregulated fraction, suggesting a clear apparent difference between the two rice genotypes in response to the saline–alkaline stress at the metabolic level, especially lipids, alkaloids, and phenolic acids (Figure 4). Lipids, as one of the main constituents of the plasma membrane, play crucial roles in plant resistance to stress responses through modulating the fluidity of the cell membranes, along with their roles in lipid-dependent signaling cascades [90,91,104]. It has been reported that lysophosphatidylcholine (lysoPC) and lysophosphatidylethanolamine (LysoPE), acting as the major components of lipids, are rapidly induced by various biotic or abiotic stress [105,106], in addition to the accumulation of unsaturated fatty acids contributing to the cell membrane fluidity [104]. In this study, lipids (19) were the largest part of the metabolites that were specifically upregulated by the saline–alkaline stress in RPY geng (saline–alkaline-tolerant genotype) (Figure 6B), such as 5 free fatty acids, 5 phosphatidylcholines (1 PC and 4 lysoPCs), and 5 LysoPEs, suggesting these lipids’ involvement in the RPY geng tolerance to saline–alkaline stress by enhancing the fluidity of the plasma membrane. Increasing reports suggest that phenolic compounds and their derivatives (collectively termed salicylic acids, ferulic acids, caffeic acids, etc.), as endogenously active secondary metabolites, play crucial roles in the plants’ defense against various abiotic and biotic stresses due to their roles in antioxidant and cell wall remodeling, together with their roles in improving the metabolic activities [107,108]. Moreover, ferulic acids and caffeic acids, also as the downstream metabolites of the TCA cycle, are involved in the process of energy metabolism [91]. In our findings, 14 phenolic acids (including salicylic acids, ferulic acids, caffeic acids, and their corresponding derivatives) were also specifically induced in the saline–alkaline-tolerant genotypes (RPY geng), suggesting that these phenolic compounds may participate in the saline–alkaline resistance of rice by maintaining ROS homeostasis and enhancing the activity of energy metabolism. Recent studies revealed that polyamines (including mainly putrescine, spermidine, and spermine) had potential roles in protecting the plants from various abiotic stresses through the modulation of hormones, ROS, membrane stability, amino acid and carbon metabolism, and ion channels [109,110]. Among them, putrescine, as a free radical scavenger, was supposed to elevate the membrane stability through altering the fatty acid level and to protect the membrane from oxidative stress damage through regulating the antioxidant system under abiotic stress [109]. Spermidine, acting as a stress-protecting compound, has been shown to alleviate plant tolerance towards various abiotic stresses through elevating the activities of the antioxidants or stabilizing the structural integrity of the photosynthetic apparatus [111,112]. Herein, nine alkaloids, consisting mostly of putrescine, spermidine, and their related derivatives, were also specifically accumulated by the saline–alkaline stress in RPY geng, suggesting that these alkaloid compounds may be involved in the saline–alkaline tolerance of rice through alleviating oxidative stress damage and stabilizing the structural integrity of the membrane or photosynthetic apparatus. Additionally, 11 other metabolites were also observed to be specifically upregulated in RPY geng under saline–alkaline stress, including 3 amino acids and their derivatives, 4 lignans, 1 coumarin, and 3 vitamins. Previous studies revealed that amino acids participated in the response to various abiotic stresses by serving as the precursor molecule for the synthesis of secondary metabolites and protein and functioning as effective antioxidants for ROS detoxification, in addition to their roles in osmotic adjustment [113]. Coumarins and lignans, as a group of phenylpropanoids, had important functions in improving the tolerance to multiple stresses for the plant’s interaction with its environment through the modulation of the cell wall, ROS scavenging capacity, or semiochemical role [114,115]. Plant-derived vitamins were essential for the metabolism in the plant’s response to its environment because of their role as enzymatic cofactors, redox chemistry, and important antioxidants [116]. Overall, based on the metabolome analysis results, it can be reasonably inferred that the capacities of osmotic potential, energy metabolism, plasma membrane fluidity, ROS scavenging, as well as total antioxidant activity, are the main factors driving the difference between RPY geng and Chao2R in response to saline–alkaline stress.

Based on comprehensive phenotype, physiology, transcriptome, and metabolome analyses of RPY geng (a saline–alkaline-tolerant genotype) and Chao2R (a saline–alkaline-sensitive genotype), we propose a hypothetical model of rice saline–alkaline adaption: the external saline–alkaline condition is firstly perceived through a complicated network of signal transduction pathways, including hormone signaling and MAPK cascade pathways; then, it activates the downstream transcription factors, thereby activating these metabolic processes consisting mainly of lipids, alkaloids, and phenolic acids by inducing the expression of the saline–alkaline-responsive genes directly, ultimately promoting saline–alkaline tolerance by increasing the capacities of osmotic potential, energy metabolism, plasma membrane fluidity, ROS scavenging, as well as total antioxidant (Figure 7).

## 4. Materials and Methods

### 4.1. Plant Materials, Growth, Treatment, and Sample Collection

The plant materials used under study include Luohui 9 (short for Chao2R, *Oryza sativa* ssp. *indica*) and RPY geng (*Oryza sativa* ssp. japonica). Seeds were firstly germinated on 1/4 MS medium containing 0.7% agar at 28 °C for 2 days. The germinated seedlings were then transferred to 96-well plate black polypropylene containers (127 mm × 87 mm × 114 mm) with Yoshida solution in the Wuhan University greenhouse under 14 h light (200 µmol m^−2^ s^−1^, 28 °C)/10 h dark (22 °C), with approximately 70% relative humidity. At 15 days after sowing (15 DAS), half the seedlings were transferred to Yoshida solution supplemented with 0.3% (*w*/*v*) mixtures of alkaline salts (Na_2_CO_3_:NaHCO_3_ = 1:3) to mimic the saline–alkaline stress conditions with a high pH value (9.35) and high Na^+^ concentration (approx. 40.84 mM) for 72 h. The remaining seedlings grown under original Yoshida solution were used as an untreated control (CK). Shoot samples were collected in the light after treatment (between 6 and 8 h), rapidly frozen in liquid N_2_, and stored at −80 °C prior to analysis. Three independent biological replicates (at least fifteen seedlings/each replicate) were applied for RNA and metabolite extractions.

To evaluate the saline–alkaline stress tolerance of Chao2R and RPY geng, 15 DAS seedlings were transferred to Yoshida solution containing 0.3% (*w*/*v*) mixtures of alkaline salts (Na_2_CO_3_:NaHCO_3_ = 1:3) for 10 days. After saline–alkaline treatment, the seedlings were cultured in the control Yoshida solution for one week recovery, and then the final survival rate of the seedlings were calculated on the two rice genotypes.

### 4.2. Measurements of Phenotypic and Physiological Traits

Images of the rice seedlings (at least 10 plants) from treatment and control groups were taken with a Nikon D7100 camera (Nikon, Tokyo, Japan), and shoot height was then analyzed as the distance from the base of the shoot of the seedling to the tip of the tallest leaf blade of each plant using ImageJ 1.50i software (National Institutes of Health, Bethesda, MD, USA). The harvested fresh rice seedlings (excluding roots) were immediately measured as shoot fresh weight. For shoot dry weight (DW) measurement, these harvested fresh samples were oven-dried for 1 h at 105 °C and for 5 days at 80 °C until they reached constant mass. Relative growth rates (RGRs) between 6 and 25 DAS were calculated as (ln(DW_n+1_) − ln(DW_n_))/(t_n+1_ − t_n_) from mean shoot biomass upon control and saline–alkaline treatments. To analyze the physiological traits of the shoot of the rice seedlings with and without saline–alkaline stress treatment, the following physiological parameters were measured according to the manufacturer’s protocols of their corresponding assay kits (Suzhou Comin Biotechnology Co., Ltd., Suzhou, China): the content of proline (Pro), soluble sugar (SS), and malondialdehyde (MDA), the activity of superoxide dismutase (SOD), peroxidase (POD), and catalase (CAT), as well as the total antioxidant capacity (T-AOC). Briefly, the proline content measurement was as follows, according to the previous study with some modification [117]: proline of shoot samples (0.1 g frozen weight) was extracted with 1 mL 3% sulfosalicylic acid at 90 °C for 10 min. After centrifugation at 10,000× *g* for 10 min, the cooling supernatant was treated with 0.5 mL acetic acid and 0.5 mL 2.5% ninhydrin solution in a boiling water bath for 30min, then 1 mL toluene was added, and the absorbance was determined at 520 nm. The soluble sugar content was performed based on the previous report with minor modifications [118]: soluble sugar content of shoot samples (0.1 g frozen weight) was extracted with 1 mL distilled water at 95 °C for 10 min, and then the cooling homogenate was centrifuged at 8000× *g* for 10 min, followed by the supernatant being filled to 10 mL with distilled water. The 0.2 mL supernatant reacted with 0.1 mL 2% anthrone solution, 1 mL 98% sulfuric acid, and 0.2 mL distilled water in a water bath at 95 °C for 10 min. After cooling, the absorbance of the mixture was measured at 620 nm. The malondialdehyde (MDA) content was measured by the thiobarbituric acid (TBA) reaction method as described by Dhindsa et al. (1981) with some modifications [119]: shoot samples (0.1 g frozen weight) were homogenized in 1 mL of 5% cold trichloroacetic acid (TCA) and centrifuged at 8000× *g* for 10 min at 4 °C. The 0.1 mL supernatant was mixed with 0.3 mL of 0.67% TBA in a water bath at 95 °C for 30 min and then immediately cooled on ice. After centrifugation at 10,000× *g* for 10 min, the absorbance of supernatant was determined at 532 nm and 600 nm, respectively. To assay the activity of antioxidant enzymes, including superoxide dismutase (SOD), peroxidase (POD), and catalase (CAT), shoot samples (0.1 g frozen weight) were homogenized in 1 mL of 50 mM cold phosphate buffer for extraction of antioxidant enzymes. After centrifugation at 8000× *g* for 10 min at 4 °C, the supernatant was subsequently used for measurement of antioxidant enzyme activity. For SOD activity assay, 90 μL supernatant reacted with 0.936 mL nitro blue tetrazolium (NBT) reaction solution for 30 min, and then the absorbance of the reaction mixture was determined at 560 nm. One unit of SOD activity was assayed by measuring its ability to inhibit 50% of NBT photoreduction at 560 nm as previously described [119]. For POD activity assay, 50 μL supernatant was added to 0.95 mL guaiacol reaction solution containing 20 mM phosphate buffer, 20 mM guaiacol, and 40 mM H_2_O_2_, and then changes of the absorbance of the reaction buffer were read at 470 nm every 1 min [120]. One unit of POD activity was defined as the amount of enzyme that would increase 0.01 absorbance unit at 470 nm min^−1^ g^−1^ fresh weight. For CAT activity assay, 35 μL supernatant was added to 1 mL reaction buffer containing 15 mM phosphate buffer and 15 mM H_2_O_2_. Reaction was initiated immediately once enzyme extract was added, followed by reading changes of the absorbance of the reaction buffer at 240 nm every 1 min [119]. One unit of CAT activity was defined as the amount of enzyme that would decrease 1nmol H_2_O_2_ min^−1^ g^−1^ fresh weight. For total antioxidant capacity (T-AOC) assay, shoot samples (0.1 g frozen weight) were homogenized in 1 mL of 50 mM cold phosphate solution. After centrifugation at 10,000× *g* for 10 min at 4 °C, 50 μL supernatant was mixed and reacted with 950 μL ferric-reducing ability of plasma (FRAP) solution for 20 min, and then the absorbance of the mixture was recorded at 593 nm using a spectrophotometer [121].

### 4.3. RNA Sequencing (RNA-seq) and Data Analysis

To investigate the transcription levels’ changes of Chao2 and RPY geng under saline–alkaline stress conditions, RNA extraction and cDNA library construction of 12 shoots of Chao2 and RPY geng from three independent biological replicates for each genotype with and without saline–alkaline stress treatment were used for RNA-seq on the BGISEQ sequencing platform by Metware Biotechnology Co., Ltd. (Wuhan, China). Total RNA was extracted from the shoots using ethanol precipitation protocol and CTAB-PBIOZOL reagent following the manufacturer’s instructions. RNA integrity was monitored using a Nano Drop and Agilent 2100 bioanalyzer (Thermo Fisher Scientific, Waltham, MA, USA). Messenger RNA (mRNA) was purified from total RNA using oligo (dT- attached magnetic beads. Short fragments of mRNA were conducted by using fragment buffer at appropriate temperature. First-strand cDNA was synthesized using the mRNA fragments as templates under random hexamer primer and M-MuLV Reverse Transcriptase (RNase H). Subsequently, second-strand cDNA synthesis was performed by using buffer, dNTPs, RNase H, and DNA polymerase I. Double-stranded cDNA were purified and processed via AMPure XP beads for end repair with addition of A-tailing mix and RNA index adapters. The cDNA fragments were purified with AMPure XP system (Beckman Coulter, Beverly, CA, USA) and enriched by PCR amplification. The final transcriptome libraries were constructed from the double-stranded PCR products, which were heated, denatured, and circularized by the splint oligo sequence, and library products’ quality was checked using the Agilent Bioanalyzer 2100 system. The single-strand circle DNA (ssCir DNA) was formatted as the final library. The final library was amplified with phi29 to make DNA nanoball (DNB) which consisted of more than 300 copies of one molecule. DNBs were loaded into the patterned nanoarray, and single-end 50-base reads were produced for sequencing analysis on BGISEQ MGISEQ-2000RS platform (BGI, Shenzhen, China). The obtained original raw data (raw reads) were cleaned using fastp v 0.19.3 by removing reads containing adapter, reads containing the ploy-N content at more than 10%, and low-quality sequences (Q ≤ 20, the number of low-quality bases contained in more than 50% in a read). The obtained clean reads were mapped to the rice reference genome available at ftp://ftp.ensemblgenomes.org/pub/plants/release-50/fasta/oryza_sativa/dna/ (accessed on 2 June 2021) using HISAT v2.1.0 tool soft, and the new genes were predicted using StringTie v1.3.4d software. In order to eliminate the influence of different genes in lengths and sequencing discrepancies, quantification of gene expression levels was estimated by the fragments per kilobase of exon per million mapped fragments (FPKM) method using featureCounts v1.6.2 software. The false discovery rate (FDR) was used as the corrected *p*–value threshold in multiple tests using the Benjamini–Hochberg method. In this study, the FDR ≤ 0.05 and |log_2_ fold change with FPKM| ≥ 1 were used as the cutoff values for significant differences in gene expression between groups using DESeq2 v1.22.1. To obtain more information of the differentially expressed genes (DEGs), the enrichment analyses consisting of Gene Ontology (GO) and Kyoto Encyclopedia of Genes and Genomes (KEGG) were carried out by following the hypergeometric test, with an FDR ≤ 0.05 as a threshold for significance.

### 4.4. Quantitative Reverse Transcription Polymerase Chain Reaction (qRT-PCR) Analysis

In order to validate the accuracy of transcriptome data, the expression of 20 selected genes, including 10 randomly selected genes and ten DEGs, were determined by using quantitative real-time reverse transcription polymerase chain reaction (qRT-PCR) assay. Primers of these 20 selected genes for qRT-PCR are listed in Appendix A. For quantification of these genes, samples of Chao2 and RPY geng used were the same as for RNA-seq. Total RNA extraction and first-strand cDNA synthesis and qRT-PCR reaction (10 µL), and the relative expression levels of 20 query genes were performed as previously described [122].

### 4.5. Metabolomics and Data Analysis

The samples for RNA-seq were next used for wide–targeted metabolic profiling using an ultra-performance liquid chromatography electrospray ionization tandem mass spectrometry (UPLC-ESI-MS/MS) (UPLC, Nexera X2, Shimadzu, Kyoto, Japan; MS, 4500 Q TRAP, Applied Biosystems, Waltham, MA, USA) platform by Metware Biotechnology Co., Ltd. according to the standard instructions. Briefly, shoot samples were freeze-dried using a vacuum freeze-dryer (Scientz-100F) and crushed into a powder using a mixer mill (MM 400, Retsch, Germany) with a zirconia bead for 1.5 min at 30 Hz. Lyophilized powder (100 mg) was weighed and extracted overnight at 4 °C with 1.2 mL of 70% methanol solution. After centrifugation at 12,000× *g* rpm for 10 min, the supernatants of the samples were filtrated using 0.22 μm pore size (ANPEL, Shanghai, China) for the next UPLC-MS/MS analysis. During the conditions of UPLC analysis, the sample effluents were alternatively connected to an ESI-triple quadrupole-linear ion trap (QTRAP)-MS. Linear ion trap (LIT) and triple quadrupole (QQQ) scans were acquired for a UPLC-MS/MS system, which was equipped with an ESI Turbo Ion-Spray interface and controlled by Analyst 1.6.3 software (AB Sciex, Framingham, MA, USA), with the operation of positive and negative ion modes. Qualitative identification of metabolites under study was obtained according to the source of public database (e.g., MassBank, KNAPSAcK, and METLIN) and private MVDB database (Metware Biotechnology Co., Ltd.). After the scaled unit variance of data, unsupervised principal component analysis (PCA) was conducted with the statistics function ‘prcomp’ in R package. The hierarchical cluster analysis (HCA) and Pearson correlation coefficients (PCCs) were both presented by the ‘pheatmap’ within R, and PCCs were calculated using the ‘cor’ function in R package. The values of variable importance in projection (VIP) were obtained from the result of orthogonal partial least squares discriminant analysis (OPLS-DA) using ‘MetaboAnalystR’ within R package. The VIP ≥ 1 and |log_2_ fold change (FC)| ≥ 1 were set as the threshold to determine significantly differentially accumulated metabolites (DAMs) between groups. To obtain more functions of identified metabolites, these metabolites were annotated with the KEGG compound database and were next mapped to the KEGG pathway database. Metabolite sets enrichment analyses (MSEAs) were carried out with a hypergeometric test’s *p*-values ≤ 0.05 as the cutoff value for significance.

### 4.6. Statistical Analyses

All data were analyzed by two-way ANOVA using SPSS17.0 or GraphPad Prism 8. Significant differences between the multiple treatment groups were evaluated using Student’s *t*-test (*p* < 0.05). All heatmaps of expression levels in the study were carried out by TBtools [123] or R package. A co-expression network of candidate genes as well as an interactive network between candidate genes and metabolites were carried out using Cytoscape v3.8.2 software by following the screening criteria of the absolute value of |PCC| ≥ 0.9 and *p* < 0.01.

## 5. Conclusions

In this study, we explored a potential molecular mechanism difference in salt–alkaline tolerance between *indica* Chao2R and *japonica* RPY geng at the seedling stage through the investigation of changes in genes and metabolites by integrating phenotype, physiology, transcriptome, and metabolome analyses. Compared with Chao2R, RPY geng is more inclined to activate these metabolic processes, including mainly lipids, alkaloids, and phenolic acids, through a series of cascade regulations of these genes mainly involved in plant signal transduction pathways (hormone signaling and MAPK cascade pathways), such as *PR*, *FLS2*, *VIP1*, *BRI1*, *AUX/IAA*, *JAZ*, and *GID1*; transcription factors, such as *NAC* and *WRKYs*; and saline–alkaline-responsive genes, such as *4CL*, *GPX*, *GST*, *MPG*, *HK*, *GAD*, *ACX*, *LOX*, and *ACS*, so as to promote saline–alkaline stress tolerance by improving the abilities of ROS scavenging, osmotic potential, energy metabolism, and plasma membrane fluidity, etc. The findings of this study will be valuable for providing a mechanistic understanding of the rice saline–alkaline tolerance at the transcriptomic and metabolomic levels.

## Figures and Tables

**Figure 1 ijms-24-12387-f001:**
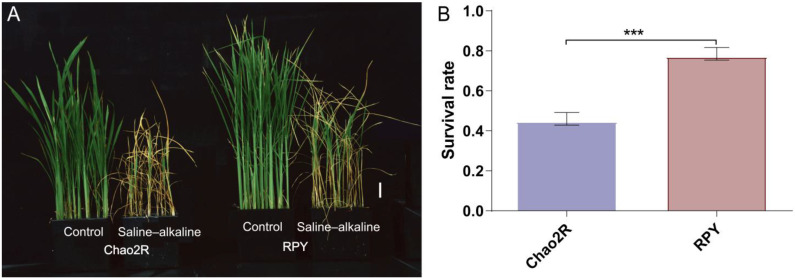
Effects of saline–alkaline stress on Chao2R and RPY geng at the seedlings. (**A**) Growth phenotype of both rice cultivars with and without saline–alkaline treatment. Scale bars = 3 cm. (**B**) The final survival rate of the two rice cultivars after 10 days of saline–alkaline treatment followed by 7 days for recovery at normal conditions. Asterisks represent significance between groups by Student’s *t*-test (*** *p* < 0.001).

**Figure 2 ijms-24-12387-f002:**
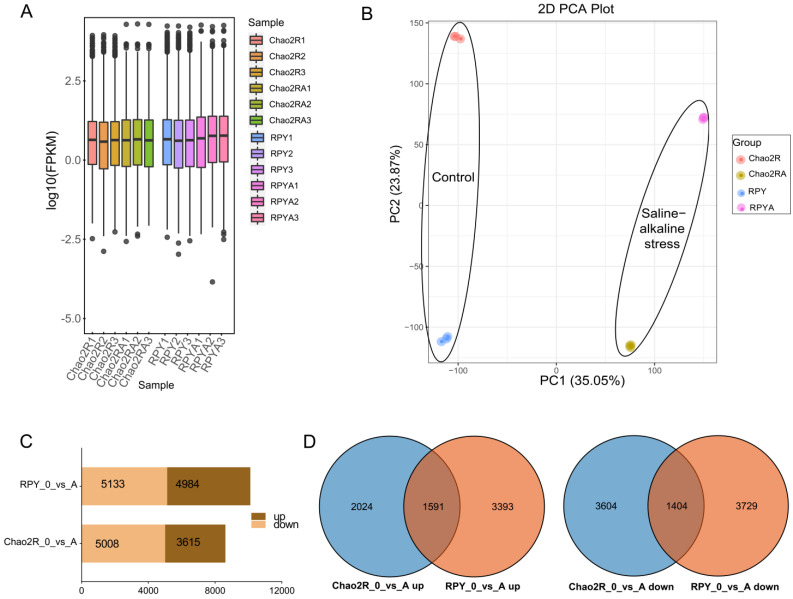
Comparative transcriptome analysis of Chao2R and RPY geng seedlings under saline–alkaline stress conditions. (**A**) FPKM box diagram of all tested 12 samples under study. Chao2R and RPY indicate samples of Chao2R and RPY geng under normal conditions, respectively; Chao2RA and RPYA indicate samples of Chao2R and RPY geng treated with saline–alkaline stress for 72 h, respectively. (**B**) Principal component analysis (PCA) of transcriptome data from the two rice cultivars with and without saline–alkaline stress. (**C**) Number of differential expression genes (DEGs) in Chao2R and RPY geng after exposure to saline–alkaline condition. 0 and A represent the untreated control (CK) and the saline–alkaline treatment, respectively. (**D**) Venn diagram of upregulated DEGs (left) and Venn diagram of downregulated DEGs (right) in Chao2R and RPY geng.

**Figure 3 ijms-24-12387-f003:**
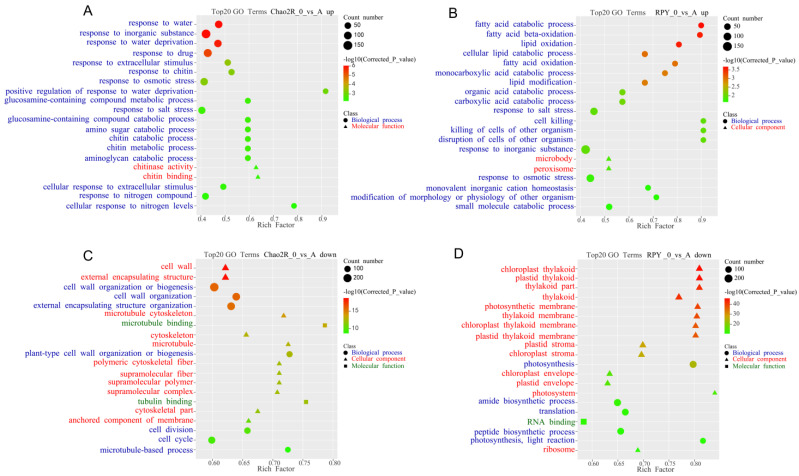
Top 20 Gene Ontology (GO) enrichment of DEGs from Chao2R and RPY geng under saline–alkaline stress for 72 h in comparison with untreated control (short for 0_vs_A). Upregulated DEGs in Chao2R (**A**), upregulated DEGs in RPY geng (**B**), downregulated DEGs in Chao2R (**C**), downregulated DEGs in RPY geng (**D**).

**Figure 4 ijms-24-12387-f004:**
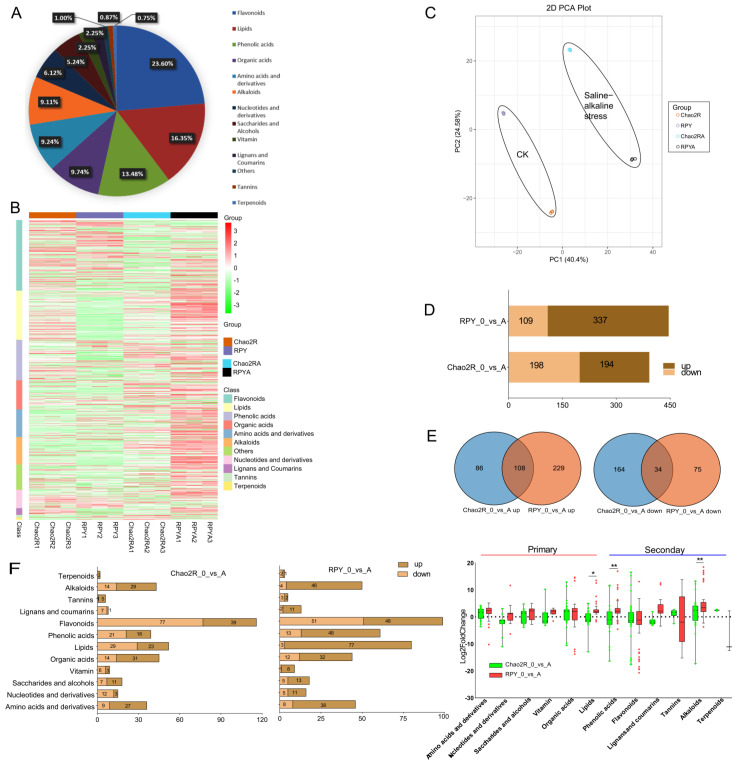
Comparative metabolomic analysis of Chao2R and RPY geng at the seedlings under saline–alkaline stress treatment. (**A**) Classification of the identified 801 metabolites under study. (**B**) Clustering heatmap of all metabolites. Chao2R and RPY indicate samples of Chao2R and RPY geng without saline–alkaline treatment, respectively; Chao2RA and RPYA indicate samples of Chao2R and RPY geng with saline–alkaline stress treatment for 72 h, respectively. (**C**) PCA of metabolome data from the two rice cultivars with and without saline–alkaline stress treatment. (**D**) Number of differentially accumulated metabolites (DAMs) in Chao2R and RPY geng after subjected to saline–alkaline condition in comparison with untreated control (short for 0_vs_A). (**E**) Venn diagram of upregulated DAMs (left) and Venn diagram of downregulated DAMs (right) in Chao2R and RPY geng. (**F**) Statistics numbers (left) and log_2_ (fold change) values (right) of DAMs distribution for different types of metabolites in Chao2R and RPY geng treated with salt–alkaline stress in comparison with untreated control. Asterisks represent significance between the groups by Student’s *t*-test (* *p* < 0.05 and ** *p* < 0.01).

**Figure 5 ijms-24-12387-f005:**
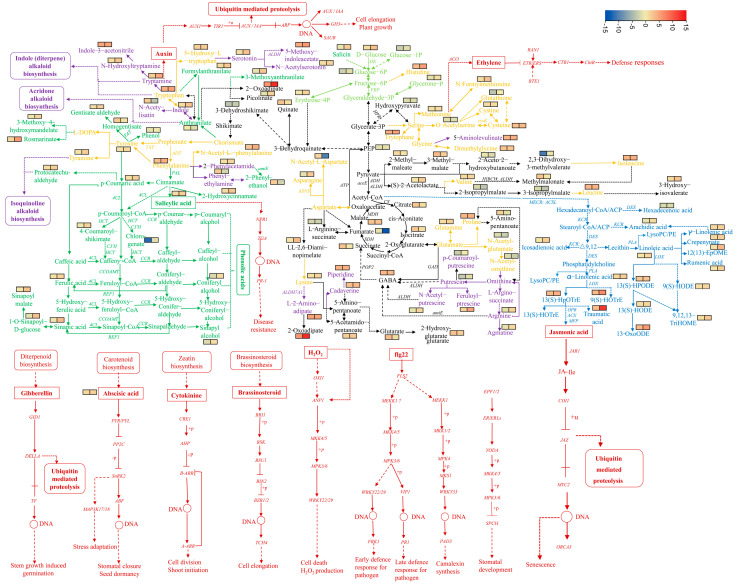
Main metabolic pathways of Chao2R and RPY geng seedlings in response to saline–alkaline stress. Metabolites and genes were mapped to an integrated systemic metabolic pathway diagram based on combining the KEGG pathways of DAMs and DEGs. The first and second boxes indicate the log_2_ (fold change) of metabolites in Chao2R and RPY geng, respectively, after subjected to saline–alkaline treatment in contrast to untreated control, in which red indicates upregulation, and blue indicates downregulation. The metabolic pathways involvement in alkaloid, phenolic acid, lipid, amino acid, organic acid, saccharide, and plant signal transduction are marked in purple, green, blue, orange, black, light green, and red, respectively.

**Figure 6 ijms-24-12387-f006:**
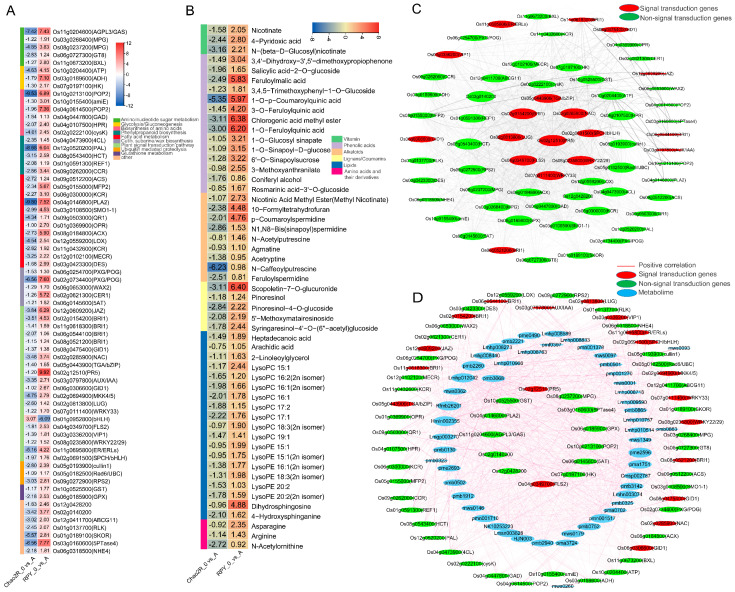
Potential important candidate genes and metabolites for rice saline–alkaline tolerance. (**A**) Expression patterns of important candidate genes. The data in the boxes indicate the log_2_ (fold change) of genes in Chao2R and RPY geng after exposure to saline–alkaline condition compared with untreated control (short for 0_vs_A). The upregulation and downregulation are masked in blue and red, respectively. (**B**) The expression heatmap showing the log_2_ (fold change) of important candidate metabolites in Chao2R and RPY geng after exposure to saline–alkaline condition in comparison with untreated control (short for 0_vs_A). Blue and red indicate downregulation and upregulation, respectively. (**C**) Co-expression network of candidate genes (Pearson correlation coefficient (PCC) ≥ 0.9, *p* < 0.01). (**D**) An interactive correlation network of potential important candidate genes and metabolites involved in rice salt–alkaline tolerance (PCC ≥ 0.9, *p* < 0.01). Red lines indicate the positive correlation, blue solid ellipses indicate metabolites, red solid ellipses indicate genes involved in plant signal transduction pathways, and green solid ellipses indicate genes related to non-signal transduction pathways.

**Figure 7 ijms-24-12387-f007:**
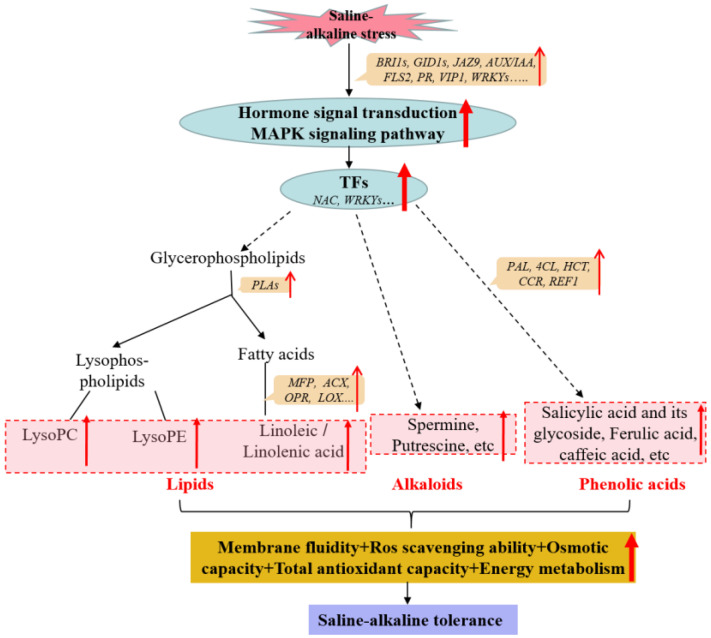
Proposed molecular models for rice saline–alkaline tolerance. Under the saline–alkaline stress, in contrast to Chao2R, RPY geng may be more inclined to active the downstream transcription factors through these genes’ cascade regulation between the plant hormone signaling transduction pathway and the plant MAPK signaling pathway, then activating these metabolic processes, including lipids, alkaloids, and phenolic acids, which ultimately improves the ability of ROS scavenging, osmotic pressure, plasma membrane fluidity, energy metabolism, and total antioxidants.

**Table 1 ijms-24-12387-t001:** Phenotypical and physiological characteristics of Chao2R and RPY geng with and without the saline–alkaline conditions.

Trait	Chao2R		RPY Geng	
	CK	0.3%A	CK	0.3%A
SFW (g)	0.235 ± 0.021 ^a^	0.198 ± 0.016 ^a^	0.456 ± 0.027 ^c^	0.308 ± 0.065 ^b^
SDW (g)	0.047 ± 0.009 ^b^	0.032 ± 0.004 ^a^	0.090 ± 0.0045 ^c^	0.051 ± 0.008 ^b^
SH (cm)	26.388 ± 1.195 ^a^	21.511 ± 1.478 ^b^	37.436 ± 0.590 ^c^	31.979 ± 1.305 ^d^
RGR (g g^−1^ d^−1^)	0.062 ± 0.007 ^a^	0.022 ± 0.003 ^b^	0.107 ± 0.002 ^c^	0.049 ± 0.004 ^d^
Pro (μg g^−1^)	114.375 ± 5.166 ^a^	617.825 ± 31.839 ^b^	113.619 ± 6.381 ^a^	525.277 ± 10.187 ^c^
SS (mg g^−1^)	9.982 ± 1.259 ^a^	11.188 ± 0.404 ^b^	10.257 ± 0.150 ^a,b^	16.574 ± 0.296 ^c^
MDA (nmol g^−1^)	2.123 ± 0.398 ^a^	4.056 ± 0.101 ^b^	2.473 ± 0.464 ^a^	3.583 ± 0.054 ^b^
SOD (U g^−1^)	1291.054 ± 30.045 ^a^	2811.185 ± 118.884 ^b^	2432.248 ± 62.447 ^c^	3492.370 ± 208.532 ^d^
POD (U g^−1^)	9787.804 ± 80.488 ^a^	12,079.593 ± 14.285 ^b^	7990.000 ± 130.000 ^c^	26,776.920 ± 84.620 ^d^
CAT (nmol min^−1^ g^−1^)	925.068 ± 121.810 ^a^	885.388 ± 58.494 ^a^	1689.188 ± 236.332 ^b^	1471.082 ± 24.087 ^b^
T-AOC (μmol Trolox g^−1^)	2.721 ± 0.039 ^a^	2.817 ± 0.067 ^a^	2.633 ± 0.044 ^a^	3.939 ± 0.097 ^b^

Data are means ± SE (*n* ≥ 3). Small letters represent significant difference (*p* < 0.05) within the same trait. CK represents the untreated control group, 0.3%A represents the saline–alkaline treatment group. SFW: Shoot fresh weight; SDW: Shoot dry weight; SH: Shoot height; RGR: Relative growth rate; Pro: Proline content; SS: Soluble sugar content; MDA: Malondialdehyde content; SOD: Superoxide dismutase; POD: Peroxidase activity; CAT: Catalase activity; T-AOC: Total antioxidant capacity.

## Data Availability

The raw data of the RNA-seq from the next generation sequence under study were deposited into the National Center for Biotechnology Information (NCBI) Sequence Reads Archive (SRA) database under accession number PRJNA961320.

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
