# Peer review of "Integrated Transcriptomic and Metabolomic Analyses Uncover the Differential Mechanism in Saline–Alkaline Tolerance between *Indica* and *Japonica* Rice at the Seedling Stage"

_ijms, 2023, doi:10.3390/ijms241512387_

Round 1

Reviewer 1 Report

In the paper entitled , Integrated Transcriptomic and Metabolomic Analyses Uncover the Differential Mechanism in Saline-Alkaline Tolerance Between Indica and Japonica Rice at the Seedling Stage authors have concluded that this study will highlight  important insights for understanding the mechanism underlying of rice saline-alkaline tolerance at the transcriptome and metabolome levels. 

Further the study will also provide candidate key target genes for further enhancing rice saline-alkaline stress tolerance.

Results indicate that 68 genes involved in metabolic pathways mainly including fatty acid, amino acid (such as phenylalanine and tryptophan), phenylpropanoid biosynthesis, energy metabolism (such as Glycolysis and TCA cycle), as well as signal transduction (such as hormone and MAPK signaling) were identified to be specifically upregulated in RPY geng under saline-alkaline condition, implying that a series of cascade changes of these genes to promote saline-alkaline stress tolerance.

In the introduction give an account of saline alkaline soils in your region and particularly rice growing soils and need for development of salt tolerance in rice. 

Already developed salt tolerance rice cultivars in previous studies and their performance under lab as well as field conditions, relation between drought salt and water stress in general osmotic stress which prevails under field conditions, 

go through whole document and improve the english language for international readership 

On discussion elaborate the DEGs and two fold increase in salt stress and the genes associated thereof 

801 metabolites studied for contrasting cultivars for salt stress , give more insight into the recent literature in this area particularly indica vs japonica rice 

candidate gene mining in rice 

phenotyping difficulties during fine mapping of candidate genes using bi parental populations in QTL mapping need to be discussed in breadth with support from recent literature

Author Response

Response to Reviewer 1 Comments

We thank the reviewers’ positive notes as well as critical but constructive comments to strengthen our manuscript. Reviewers’ comments are shown with a black font. Our responses are marked by red font.

Comments and Suggestions for Authors

In the paper entitled, Integrated Transcriptomic and Metabolomic Analyses Uncover the Differential Mechanism in Saline-Alkaline Tolerance Between Indica and Japonica Rice at the Seedling Stage authors have concluded that this study will highlight important insights for understanding the mechanism underlying of rice saline-alkaline tolerance at the transcriptome and metabolome levels. 

Further the study will also provide candidate key target genes for further enhancing rice saline-alkaline stress tolerance.

Results indicate that 68 genes involved in metabolic pathways mainly including fatty acid, amino acid (such as phenylalanine and tryptophan), phenylpropanoid biosynthesis, energy metabolism (such as Glycolysis and TCA cycle), as well as signal transduction (such as hormone and MAPK signaling) were identified to be specifically upregulated in RPY geng under saline-alkaline condition, implying that a series of cascade changes of these genes to promote saline-alkaline stress tolerance.

Point 1: In the introduction give an account of saline alkaline soils in your region and particularly rice growing soils and need for development of salt tolerance in rice. 

Response 1: Thanks very much for your comments. Given your suggestion, in the introduction we have provided the account of saline alkaline soils and the data of rice planting area in saline alkaline soils in China.

Point 2: Already developed salt tolerance rice cultivars in previous studies and their performance under lab as well as field conditions, relation between drought salt and water stress in general osmotic stress which prevails under field conditions, 

Response 2: As suggested by the Reviewer, we have given an account of the relation between drought salt and water stress in general osmotic stress from the recent literature in this area in the introduction.

Point 3: go through whole document and improve the English language for international readership 

Response 3: We agree with this suggestion from reviewer. As Reviewer suggested, we have now worked on both language and readability throughout the text as appropriate. We really hope that the language level have been substantially improved.

Point 4: On discussion elaborate the DEGs and two fold increase in salt stress and the genes associated thereof 801 metabolites studied for contrasting cultivars for salt stress, give more insight into the recent literature in this area particularly indica vs japonica rice candidate gene mining in rice, phenotyping difficulties during fine mapping of candidate genes using bi parental populations in QTL mapping need to be discussed in breadth with support from recent literature

Response 4: Thank you very much for the important suggestions to strengthen our manuscript. To address your comments, we have now added the recent literature of indica vs japonica rice candidate gene mining in this area, and we have also discussed elaborately regarding phenotyping difficulties during fine mapping of candidate genes using bi parental populations in QTL mapping. To improve the accuracy and breadth of major QTL loci and candidate genes associated with saline-alkaline tolerance in japonica ‘RPY geng’, it is expected to combine the two methods of bi-parental QTL mapping and GWAS for analyzing the genetic structure of saline-alkaline tolerant trait in rice by using 272 RILs linkage mapping populations and natural geographically diverse population in the discussion.   

Thank you very much again for your valuable suggestions and comments to improve our paper.

Reviewer 2 Report

This is an interesting study and the authors have collected a unique dataset using cutting-edge methodology. The paper is generally well-written and structured. However, in my opinion, the paper has some shortcomings in regard to figure quality and text.

Also, the grammar needs to be cross-checked.

Please focus on these things carefully

There is little need for English correction.

Author Response

Response to Reviewer 2 Comments

We express our sincere gratitude for your critical but constructive comments to improve our manuscript. Reviewers comments are shown with a black font. Our responses are marked by red font.

Comments and Suggestions for Authors

Point 1: This is an interesting study and the authors have collected a unique dataset using cutting-edge methodology. The paper is generally well-written and structured. However, in my opinion, the paper has some shortcomings in regard to figure quality and text.

Response 1: Thank you very much for your comments and affirmation on our paper. In accordance with your suggestion, we have increased a higher font size and resolution of the figure 2, 3 and 4 for the readability. In additon, all figures reformatted and increased a larger resolution in the text are resubmitted with Figures_Graphics_ Images material.

Point 2: Also, the grammar needs to be cross-checked.

Please focus on these things carefully

Response 2: Thanks very much for your comments. We agree with this suggestion from reviewer. Considering your suggestion, we have now revised the grammar going through the whole text as appropriate. We really hope that the grammar level have been substantially improved, and we also hope our rivisions are satisfactory to you.

Point 3: Comments on the Quality of English Language

There is little need for English correction

Response 3: We all thank your comments on the quality of English language. We have now worked on both language and readability going throughout the whole maniscript as appropriate. We hope that the language level have been substantially improved more.

Thank you very much again for your valuable suggestions and comments for improving our paper.

Reviewer 3 Report

Comments and Suggestion for Authors

Integrated Transcriptomic and Metabolomic Analyses Uncover the Differential Mechanism in Saline-Alkaline Tolerance Between Indica and Japonica Rice at the Seedling Stage

The subject of the work is very interesting and relevant since climatic alterations are already being responsible more frequent extreme events, as saline periods, which negative influences the growth of crops.

In introduction, please add some line of global effects of saline stress and percentage of saline area locally and worldwide.

Line No 66. Please correct the reference no. 14 reference is missing.

Line No 125. Please use another word instead of this Elucidate.

Line No 152. In table 1 please add the P value of all the treatments.

Line No 813. In material and methods section if this fresh weight or frozen weight, please correct this.

Line No 930. In Statistical Analyses section you can use one way ANOVA, but in this manuscript, there are 2 varieties for analysis. How you justify the one-way ANOVA for this experiment.

The English language must be require significant improvement. 

Author Response

Response to Reviewer 3 Comments

We express our sincere gratitude for your critical but constructive comments to improve our manuscript. Reviewers’ comments are shown with a black font. Our responses are marked by red font.

Comments and Suggestions for Authors

Integrated Transcriptomic and Metabolomic Analyses Uncover the Differential Mechanism in Saline-Alkaline Tolerance Between Indica and Japonica Rice at the Seedling Stage

The subject of the work is very interesting and relevant since climatic alterations are already being responsible more frequent extreme events, as saline periods, which negative influences the growth of crops.

Point 1: In introduction, please add some line of global effects of saline stress and percentage of saline area locally and worldwide.

Response 1: Thanks very much for your comments. Given your suggestion, we have added the account of global effects on saline alkaline soils, and percentage data of saline alkaline area locally and worldwide is also provided in the introduction.

Point 2: Line No 66. Please correct the reference no. 14 reference is missing.

Response 2: Thanks for your suggestions. To address your comments, we have added the missing no. 14 reference into the revised text.

Point 3: Line No 125. Please use another word instead of this Elucidate.

Response 3: We agree with the reviewer’s comment, and we have made correction the previous word of “elucidate“ to “explore” according to the Reviewer’s comments. In addition, word “elucidate” used in the similar condition has also been changed in the revised text.

Point 4: Line No 152. In table 1 please add the P value of all the treatments.

Response 4: Thanks very much for your comments. We agree with the reviewer’s comment that the P value should be added in all the treatments. Two-way ANOVA were selected to analyze the data using multiple comparisons within two varieties without the control group, it was not convenient to add P value in table 1. To improve the accuracy of data analysis, the P value of all the treatments are shown by using additional supplementary Figure S1 in the revised text. We really hope our revisions are satisfactory to your expectation.

Point 5: Line No 813. In material and methods section if this fresh weight or frozen weight, please correct this.

Response 5: Thanks for your comments to improve the accuracy of our paper. As Reviewer suggested, we have corrected “fresh weight” to “frozen weight” in the material and method section.

Point 6: Line No 930. In Statistical Analyses section you can use one way ANOVA, but in this manuscript, there are 2 varieties for analysis. How you justify the one-way ANOVA for this experiment.

Response 6: Thanks for your suggestions and pointing out the error in our paper. Two-way ANOVA instead of one-way ANOVA were used to analyze for this experiment during the data process. We have therefore corrected “one-way ANOVA” to “two-way ANOVA” in the revised manuscript.

Point 7: Comments on the Quality of English Language

The English language must be require significant improvement. 

Response 7: We agree with this suggestion from reviewer. As Reviewer suggested, we have now worked on both language and readability going throughout the whole text as appropriate. We really hope the language level in the revised text have been substantially improved, and we also hope our revisions are satisfactory to your request on the quality of English language.

To address your comments, we have carefully revised the whole manuscript again and again to avoid the similar problem again in the new manuscript. We are again special grateful to you for suggesting these revisions to improve our paper.
